# Therapeutic Mechanisms and Clinical Effects of Glucagon-like Peptide 1 Receptor Agonists in Nonalcoholic Fatty Liver Disease

**DOI:** 10.3390/ijms24119324

**Published:** 2023-05-26

**Authors:** Han Ah Lee, Hwi Young Kim

**Affiliations:** Department of Internal Medicine, College of Medicine, Ewha Womans University, Seoul 07985, Republic of Korea; amelia86@naver.com

**Keywords:** nonalcoholic steatohepatitis, diabetes, liraglutide, semaglutide, tirzepatide

## Abstract

Nonalcoholic fatty liver disease (NAFLD) can lead to liver fibrosis and cirrhosis. Recently, glucagon-like peptide 1 receptor agonists (GLP-1RAs), a class of drugs used to treat type 2 diabetes and obesity, have shown therapeutic effects against NAFLD. In addition to reducing blood glucose levels and body weight, GLP-1RAs are effective in improving the clinical, biochemical, and histological markers of hepatic steatosis, inflammation, and fibrosis in patients with NAFLD. Additionally, GLP-1RAs have a good safety profile with minor side effects, such as nausea and vomiting. Overall, GLP-1RAs show promise as a potential treatment for NAFLD, and further studies are required to determine their long-term safety and efficacy.

## 1. Introduction

Nonalcoholic fatty liver disease (NAFLD) is characterized by the accumulation of fat in the liver, which can progress to nonalcoholic steatohepatitis (NASH), fibrosis, and cirrhosis [1]. Therefore, the appropriate management of NAFLD is imperative in preventing the progression of the disease and mitigating the risk of developing serious complications. NAFLD is commonly associated with obesity, insulin resistance, and metabolic syndrome [2,3]. Antidiabetic medications improve insulin resistance, decrease inflammation, and reduce liver fat accumulation, which are key factors in the pathogenesis of NAFLD. Therefore, antidiabetics have been extensively studied for their potential role in the management of NAFLD [4,5,6].

Glucagon-like peptide-1 receptor agonists (GLP-1RAs) are a class of medications commonly used in the treatment of type 2 diabetes mellitus (T2DM) [7]. GLP-1 is a peptide hormone produced by gut enteroendocrine cells that affects the secretion of insulin and glucagon, food intake, and gut motility, and maintains postprandial glucose homeostasis [8]. GLP-1RAs function by stimulating the GLP-1 receptor, which is expressed in various organs, including the pancreas, liver, and gastrointestinal tract [9]. This results in increased insulin secretion, decreased glucagon secretion, and delayed gastric emptying, thereby improving glycemic control [10,11].

The strong correlation between NAFLD and metabolic disorders, such as obesity and T2DM, suggests that targeting the GLP-1 may be a promising therapeutic strategy for NAFLD. Recently, several studies have shown that GLP-1RAs may also have beneficial effects in the management of NAFLD [12,13]. Several animal models and human trials have demonstrated that GLP-1RAs can reduce hepatic steatosis, inflammation, and fibrosis [14,15,16]. The role of GLP-1RAs on NAFLD could be also clarified by its effects on body weight reduction, oxidative stress, and the gut-liver axis [12,17]. In addition, GLP-1/glucose-dependent insulinotropic peptide (GIP)-1 dual agonists are a relatively new class of medications that have shown promise in NAFLD treatment [18,19].

However, further research is required to fully elucidate their mechanism of action and determine their long-term efficacy and safety. Furthermore, it is unclear whether the treatment effects of GLP-1RAs on NAFLD are present in both patients with and without T2DM. Therefore, this narrative review will discuss the therapeutic potential of GLP-1RAs in NAFLD treatment.

## 2. Treatment Mechanisms of GLP-1RAs on NAFLD

The treatment mechanism of GLP-1RAs in NAFLD involves various pathways (Figure 1). The main function of GLP-1 is that it activates insulin secretion from beta cells and inhibits glucagon secretion from alpha cells [20]. By increasing the sensitivity of cells to insulin, GLP-1RAs help the body to regulate blood sugar levels more effectively, which can reduce the risk of developing NAFLD [21,22] (Figure 2). Additionally, GLP-1 can slow down the process of gastric emptying and gastrointestinal movement, which consequently lowers glucose absorption, regulating the postprandial glucose concentration. GLP-1 also induces satiety through direct stimulation of the central nervous system [23].

Additionally, GLP-1RAs decrease fat accumulation in the liver [22,24]. Several studies of animal models with NASH have demonstrated the use of GLP-1RAs to improve liver histology. A 4-week liraglutide treatment to mice fed high-sucrose corn syrup induced significant reductions in insulin resistance, body and liver weight, and prevented the development of either NAFLD or NASH, unlike in the control group [22]. Similarly, in mice with insulin resistance and NAFLD, an 8-week liraglutide treatment reduced liver fat content (LFC) [24]. In another study with *ob*/*ob* mice, a 60-day exedin-4 treatment significantly decreased blood glucose levels, insulin resistance, weight gain, and LFC [25]. In an in vitro study, hepatocytes with exedin-4 treatment reduced the expression of sterol regulatory element-binding transcription factor (SREBP)-1c and stearoyl-CoA desaturase (SCD)-1, which are important regulators of de novo lipogenesis [21]. Additionally, the treatment increased the expression of peroxisome proliferator-activated receptor (PPAR)-α, which is an important element in the beta-oxidation of free fatty acids.

In preclinical models of NASH, the administration of AC3174, an analogue of exenatide, led to significant reductions in body weight, LFC, and levels of plasma alanine aminotransferase (ALT) and triglyceride [24]. Additionally, AC3174 treatment was associated with decreased collagen-1 protein expression in the liver, indicating a potential improvement in liver fibrosis. In contrast to mice that were administered AC3174, mice that had similar reductions in body weight due to calorie restriction showed marginal reductions in LFC, suggesting that the decrease in LFC achieved by GLP-1RAs was not fully dependent on weight loss. Similar findings were observed in another study; a 4-week exendin-4 treatment in a high-fat-diet-induced mouse model of obesity decreased hepatic triacylglycerol content, with no significant difference in body weight or body fat [26]. This could be explained by the GLP-1RA-induced increase in phosphorylation of cyclic adenosine monophosphate activated protein kinase (AMPK), which is a suppressor of lipogenesis [26].

The cross-talk between GLP-1 receptors in the liver and adipose tissue may play an important role in the prevention of hepatic steatosis. GLP-1 receptors are present in both the liver and adipose tissue, and activation of these receptors by GLP-1 agonists can lead to a reduction in adipose tissue inflammation and an increase in adiponectin secretion, and both can, in turn, have a positive impact on the liver [27]. A reduction in inflammation in adipose tissue may decrease the release of pro-inflammatory cytokines into the bloodstream, which can lead to improved liver function. An increased adiponectin secretion may enhance insulin sensitivity and reduce hepatic lipid accumulation [28].

GLP-1RAs can also reduce inflammation and oxidative stress, both of which are involved in NAFLD progression, thereby preventing or delaying liver damage [29]. In an in vitro study, exendin-4 treatment in fat-loaded human hepatocytes appeared to protect the hepatocytes from fatty-acid-related death by inhibiting a dysfunctional endoplasmic reticulum stress response and reducing fatty acid accumulation [29]. Additionally, exendin-4 in fat-loaded hepatocytes promoted the expression of genes, such as *Beclin-1* and *LC3B-II*, which are markers of macroautophagy. Similar results were observed in mouse liver treated with liraglutide. In studies with diet-induced NAFLD mice, GLP-1RAs treatment was effective in improving LFC, which is attributed to a reduction in oxidative stress achieved through a specific cellular pathway involving sirtuin 1 [30,31].

In addition, hyperglycemia activates the carbohydrate-responsive element-binding protein (ChREBP) transcriptional factor, a key modulator of de novo liver lipogenesis, suggesting that GLP-1RAs could reduce ChREBP activation and, consequently, decrease de novo liver lipogenesis and LFC [32]. The potential benefits of GLP-1RAs in lipid metabolism have been demonstrated in experimental studies; GLP-1RAs reduced very-low-density lipoprotein (VLDL) production and the hepatic expression of genes involved in VLDL production and lipogenesis, such as SREBP-1c and fatty acid synthase [33]. This evidence confirms that GLP-1RAs reduce the accumulation of hepatic triglycerides by limiting hepatic lipogenesis.

It is still unclear whether the effect of GLP-1RAs is a result of the direct activation of the hepatic GLP-1 receptor, or an indirect effect caused by weight loss, glycemic control, changes in lipoprotein metabolism, or improved insulin sensitivity [15,34]. Thus far, the presence of GLP-1R in liver cells is controversial [32,35].

## 3. Beneficial Effects of GLP-1RAs in Patients with NAFLD

In T2DM, GLP-1RAs are generally recommended as a second-line therapy after metformin, along with other drug classes, such as sodium-glucose co-transporter 2 (SGLT2) and dipeptidyl peptidase-4 (DPP-4) inhibitors [36]. GLP-1RAs are currently available as twice-daily (exenatide), once-daily (liraglutide and lixisenatide), or once-weekly subcutaneous injections (semaglutide, dulaglutide, and exenatide extended-release) and as a once-daily tablet for oral administration (oral semaglutide) (Table 1) [37].

Several phase II trials and investigator-sponsored studies have explored the potential of GLP-1RAs in treating NAFLD and NASH (Table 2).

### 3.1. Liraglutide

In the LEAN study, the treatment efficacy and safety of the once-daily liraglutide (1.8 mg) were compared with those of the placebo after 48 weeks in 52 overweight patients with clinical evidence of NASH [12]. The primary endpoint, NASH resolution without worsening fibrosis, was achieved in 39% of the liraglutide group, compared with 9% of the placebo group (relative risk [RR] = 4.3; 95% CI, 1.0–17.7; *p* = 0.019). A progression of fibrosis was observed in 9% and 36% of patients in the liraglutide and placebo group, respectively (RR = 0.2; 95% CI, 0.1–1.0; *p* = 0.04). The primary outcome was achieved in 38% of patients with T2DM and 40% without. The RR for achieving the primary outcome in the liraglutide group compared with the placebo was 4.7 (95% CI, 0.3–75.0; *p* = 0.20) in patients with T2DM, and 3.4 (95% CI, 0.8–14.4; *p* = 0.11) in patients without T2DM. The patients who underwent liraglutide treatment showed significant improvement in body weight, body mass index, and HbA1c and γ-glutamyl transferase (GGT) concentrations compared to the placebo, whereas no difference was observed in the change in serum aminotransferase concentrations. In a substudy of the LEAN trial that included 14 patients, liraglutide significantly reduced low-density lipoprotein (LDL)-cholesterol, ALT, serum leptin, adiponectin, and CCL-2 concentrations (all *p* < 0.05). In addition, a significant increase in adipose tissue insulin sensitivity, and decrease in hepatic de novo lipogenesis, was observed in the liraglutide group (all *p* < 0.05) [16]. The majority of adverse events were mild to moderate in severity, and similar occurrence rates between the two groups across all organ classes and symptoms were observed. However, gastrointestinal disorders were reported more frequently in the liraglutide group (81%) than in the placebo group (65%).

In other studies, the effect of liraglutide treatment for hepatic steatosis in patients with NAFLD with and without T2DM were investigated using a magnetic resonance imaging (MRI)-based assessment. In a randomized controlled trial (RCT) of patients with T2DM and NAFLD under inadequate glycemic control using metformin, the treatment effect of liraglutide, sitagliptin, and insulin glargine for 26 weeks were compared [38]. The treatment with once-daily liraglutide (1.8 mg) for 26 weeks significantly decreased MRI-proton density fat fraction (PDFF) (15.4% ± 5.6% to 12.5% ± 6.4%, *p* < 0.001), and changes from the baseline in the MRI-PDFF were significantly greater with liraglutide than with insulin glargine; however, these changes did not differ significantly between liraglutide and sitagliptin. In another RCT that compared the treatment efficacy of once-daily liraglutide (3.0 mg) and lifestyle modification (LSM) for 26 weeks in non-diabetic patients with obesity and NAFLD, similar reductions in the LFC measured using an MRI were observed between the liraglutide (−8.1 ± 13.2%) and LSM (−7.0 ± 7.1%) groups (*p* = 0.78) [44]. However, during the follow-up period without active intervention, the liraglutide group regained weight (1.8 ± 2.1 kg) and LFC (4.0 ± 5.3%), whereas those were unchanged in the LSM group.

### 3.2. Semaglutide

Semaglutide is another GLP-1RA approved for T2DM treatment. Semaglutide has a similar mechanism of action to that of liraglutide, having greater metabolic effects [53]. The efficacy of once-daily subcutaneous semaglutide compared to the placebo was investigated in 320 patients with biopsy-confirmed NASH and liver fibrosis (stage F1, F2, or F3), and patients with cirrhosis were excluded [13]. Among the patients with stage F2 or F3 fibrosis, the primary endpoint (i.e., NASH resolution without worsening of fibrosis) after 72 weeks was achieved in 40% with 0.1 mg semaglutide, 36% with 0.2 mg semaglutide, and 59% with 0.4 mg semaglutide, compared with 17% in the placebo group (*p* < 0.001 for 0.4 mg semaglutide vs. placebo); thus, this indicates its superior efficacy to that of liraglutide in the LEAN study [12]. The percentage of patients who achieved the secondary outcome (i.e., improvement in liver fibrosis stage without worsening of NASH) was comparable between the two groups (43% with 0.4 mg semaglutide and 33% with the placebo; *p* = 0.48); in contrast, fibrosis worsened in 5% of patients in the 0.4 mg semaglutide group, compared with 19% in the placebo group. In addition, semaglutide resulted in dose-dependent body weight reduction, with mean changes of −5%, −9%, −13%, and −1% in the 0.1 mg semaglutide, 0.2 mg, 0.4 mg, and placebo group, respectively. Dose-dependent reductions in liver enzyme levels and exploratory biomarker levels were also observed with semaglutide. In both patients with or without T2DM, dose-dependent reductions in HbA1c levels were observed. Gastrointestinal adverse events were more frequent in the 0.4 mg semaglutide group than in the placebo group. Among patients who discontinued treatment because of adverse events (7% of the semaglutide group [all doses] and 5% of the placebo group), gastrointestinal disorders were the most common reasons for discontinuation in the semaglutide group (4%), unlike the placebo group. Although the incidence of hepatic events was comparable between all groups, the semaglutide groups (5−7%) experienced gallbladder-related disorders more frequently than the placebo group (2%). A higher percentage of patients in the semaglutide groups had higher amylase and lipase levels than those in the placebo group, albeit with no overt case of acute pancreatitis [12].

In a phase I study, the treatment effects of semaglutide in NAFLD were investigated using MRI methods [46]. Sixty-seven patients with liver stiffness of 2.50–4.63 kPa, measured using magnetic resonance elastography, and liver steatosis ≥10%, measured using MRI-PDFF, were included. The primary endpoint (change in liver stiffness at week 48) was comparable between 0.4 mg semaglutide once-daily and the placebo (estimated treatment ratio 0.96; 95% CI, 0.89–1.03; *p* = 0.2798). No significant differences in liver stiffness were observed at weeks 24 or 72. However, reductions in liver steatosis were significantly greater with semaglutide (estimated treatment ratios: 0.70 [95% CI, 0.59–0.84; *p* = 0.0002], 0.47 [95% CI, 0.36–0.60; *p* < 0.0001], and 0.50 [95% CI, 0.39–0.66; *p* < 0.0001]), and more patients achieved a ≥30% reduction in LFC with semaglutide than with the placebo at weeks 24, 48, and 72 (all *p* < 0.0001). Significant differences between semaglutide and the placebo in visceral adipose tissue (VAT) and abdominal subcutaneous adipose tissue (SAT) were also observed at all timepoints (all *p* < 0.0001). Semaglutide induced significant reductions in body weight compared with the placebo at all timepoints (estimated treatment difference of −9.68%; 95% CI, –12.58 to 6.77; *p* < 0.0001 at week 72). In patients with T2DM, mean changes in HbA1c level were significantly greater with semaglutide than in those with the placebo at all time points (*p* < 0.0001 at weeks 24, 48, and 72). Changes in fasting glucose levels were significantly greater in the semaglutide group (*p* < 0.05). However, no significant difference was observed in HOMA-IR with semaglutide at all time points. Reductions in liver enzymes were significantly greater with semaglutide compared to with the placebo at weeks 48 and 72 (*p* < 0.05). The adverse event profiles were comparable between the two groups (93.9% vs. 87.9%). Gastrointestinal disorders were more frequently observed with semaglutide than with the placebo. Serious adverse events occurred in 12.1% and 9.1% of the semaglutide and placebo group, respectively. In patients with T2DM, hypoglycemic events developed in 14.8% of the semaglutide group, compared with 9.5% of the placebo group.

Given that the phase II trial of semaglutide in NASH treatment did not include patients with cirrhosis, its efficacy and safety in NASH-related cirrhosis are unknown. Recently, a multicenter phase II trial investigated the efficacy and safety of semaglutide in patients with obesity and biopsy-confirmed NASH-related compensated cirrhosis [47]. After 48 weeks, there was no significant difference in the proportion of patients with improvement in liver fibrosis of one or more stages without a worsening of NASH between the semaglutide and placebo group (11% vs. 29%, odds ratio [OR] 0.28 [95% CI, 0.06–1.24; *p* = 0.087]). In addition, NASH resolution was comparable between the two groups [34% in the semaglutide vs. 21% in the placebo (OR = 1.97; 95% CI, 0.56–7.91; *p* = 0.29)]. Changes in body weight, and concentrations of AST, ALT, triglycerides, and VLDL from the baseline, were significantly greater in the semaglutide group than that in the placebo group (all *p* < 0.05). A decrease in HbA1c levels in patients with T2DM was observed only in the semaglutide group (−1.39% vs. +0·24% [placebo]; estimated treatment differences = −1.63; 95% CI, −2.20 to 1.06; *p* = 0.001]. Safety profiles were similar between semaglutide and the placebo, and the most common adverse event was gastrointestinal disorder. Although this was the first study on GLP-1RA in patients with NASH-related cirrhosis, the efficacy was not significant, despite acceptable safety.

### 3.3. Exenatide

The effect of exenatide treatment in NAFLD has been evaluated only using MRI assessments. In a study of 76 patients with T2DM (HbA1c between 7% and 10% without antidiabetics) and NAFLD (LFC measured using magnetic resonance spectroscopy > 10%), the 24-week effect of treatment with 10 μg exenatide and insulin glargine twice daily was compared [48]. Both treatments significantly improved LFC (exenatide, 17.55 ± 12.93% [*p* < 0.001]; insulin glargine, –10.49 ± 11.38% [*p* < 0.05]), with no statistical difference between the two groups (*p* = 0.1248). Fibrosis-4 index (FIB-4) decreased marginally in the exenatide group (0.10 ± 0.26, *p* = 0.0448); however, it did not decrease in the insulin glargine group (0.13 ± 1.10, *p* = 0.8475), showing no statistical difference between the two groups (*p* = 0.2149). Exenatide induced greater reductions in AST, ALT, and GGT levels than insulin glargine (*p* < 0.05). Moreover, greater improvements in body weight, waist circumference, postprandial plasma glucose, and LDL-cholesterol were observed in the exenatide group compared to those in the insulin glargine group (all *p* < 0.05). No severe hypoglycemia or drug-related serious adverse events occurred.

Another study also showed that 10 μg exenatide twice daily induced a significant reduction in hepatic triglyceride content (23.8 ± 9.5%), compared with the reference treatment (+12.5 ± 9.6%; *p* = 0.007). However, patients with obesity and uncontrolled T2DM were included in this study, and NAFLD status at baseline was not defined [49].

### 3.4. Dulaglutide

Dulaglutide has been approved for the treatment of glycemic control in patients with T2DM, having better patient tolerability than other GLP-1RAs [54,55]. The effect of dulaglutide on a liver fat (D-LIFT) trial included 64 patients with T2DM and MRI-PDFF ≥ 6.0% at baseline [51]. After 24 weeks, a 1.5-mg dulaglutide treatment resulted in a greater reduction in LFC compared with the control (−32.1% vs. −5.7%; mean difference = −26.4% [95% CI, −44.2 to −8.6]; *p* = 0.004). The absolute change in liver stiffness measured using transient elastography was comparable between the two groups (−1.43 kPa vs. −0.12 kPa; mean difference = −1.31% [95% CI, −2.99 to 0.37]; *p* = 0.123). In addition, dulaglutide induced a significant reduction in GGT (−19.4 IU/L in dulaglutide vs. −6.3 IU/L in the placebo; mean between-group difference = −13.1 IU/L [95% CI, −24.4 to 1.8]; *p* = 0.025), whereas no reduction was observed in AST (*p* = 0.075) or ALT (*p* = 0.10). No serious drug-related adverse events were observed. Dulaglutide could be considered for the early treatment of NAFLD in patients with T2DM; however, larger RCTs are required for the validation of these results.

## 4. Meta-Analysis

Several meta-analyses regarding the effect of GLP-1RAs treatment in patients with NAFLD have been reported. A systematic review and meta-analysis evaluated the effects of GLP-1RAs on histopathologic changes in patients with NASH using two RCTs (one liraglutide and one semaglutide) [56]. GLP-1RAs showed higher efficacy on the improvement in NASH than the placebo, in terms of NASH resolution without fibrosis worsening (RR = 2.67; 95% CI, 1.87–3.81), improvement in hepatic fibrosis by at least 1 point (RR = 1.29; 95% CI, 0.99–1.70), and improvement in lobular inflammation (RR = 1.44; 95% CI, 1.11–1.86).

Another systemic review and meta-analysis of eight RCTs involving 468 patients evaluated the efficacy of GLP-1RAs in patients with T2DM and NAFLD [57]. GLP-1RAs treatment significantly decreased the LFC measured using MRI (weight mean difference [WMD] = −3.01%; 95% CI, −4.75 to −1.28; *p* = 0.007), however, it did not decrease FIB-4 (WMD = −0.04; 95% CI, −0.15 to 0.08; *p* = 0.50) compared to the control. Additionally, SAT (WMD = −28.53; 95% CI, −68.09 to −26.31; *p* < 0.00001), VAT (WMD = −29.05%; 95% CI, −42.90 to −15.9; *p* < 0.0001), ALT (WMD = −3.82 IU/L; 95% CI, −7.04 to −0.60; *p* = 0.02), AST (WMD = −2.4 IU/L; 95% CI, −4.55 to −0.25; *p* = 0.03), body weight (WMD = −3.48; 95% CI, −4.58 to −2.37; *p* < 0.0001), fasting blood glucose (WMD = −0.35 mg/dL; 95% CI, −0.06 to −0.05; *p* = 0.02), HbA1c (WMD = −0.39; 95% CI, −0.56 to −0.22; *p* < 0.00001), HOMA-IR (WMD = −1.51; 95% CI, −0.87 to −0.16; *p* = 0.005), total cholesterol (WMD = −0.31; 95% CI, −0.48 to 0.13; *p* = 0.0008), and triglycerides (WMD = −0.27; 95% CI, −0.43 to −0.11; *p* = 0.0008) increased in comparison with the control.

A recent systematic review and network meta-analysis study, including nine RCTs, compared the effects of GLP-1RAs, pioglizatone, and vitamin E treatments [58]. The administration of GLP-1RAs was associated with improved liver steatosis (OR = 4.11; 95% CI, 2.83–5.96), ballooning necrosis (OR = 3.07; 95% CI, 2.14–4.41), lobular inflammation (OR = 1.86; 95% CI, 1.29–2.68), and fibrosis (OR = 1.52; 95% CI, 1.06–2.20) compared with the placebo; thus, this showed a similar effect with pioglitazone and vitamin E.

The efficacy of GLP-1RAs was compared with other antidiabetics in a network meta-analysis of 49 trials that included 3836 patients [59]. The SGLT2 inhibitor (WMD = −6.09; 95% CI, −10.50 to −1.68) was most effective in reducing LFC measured using MRI, followed by GLP-1RA (WMD = −5.55; 95% CI, −10.40–−0.69); both were superior to other antidiabetics or the placebo in reducing LFC measured using MRI. In contrast, thiazolidinedione and the DPP4-inhibitor were more effective in the reduction in ALT and AST levels than SGLT2 inhibitors and GLP-1RAs. Lastly, GLP-1RA was most effective in reducing GGT. However, the study primarily assessed the efficacy of antidiabetics in improving liver enzymes and LFC measured using MRI, whereas the effects of these treatments on liver histological changes were not evaluated. In addition, the number of patients included in the final analysis for each outcome ranged from 30 to 835. Therefore, head-to-head studies are required to provide more solid evidence for clinical decision making.

## 5. Dual GIP and GLP-1 Receptor Agonist

Tirzepatide is a synthetic peptide composed of 39 amino acids that exhibits agonist activity at both GIP and GLP-1 receptors [60]. As it has a half-life of approximately 5 d, tirzepatide can be administered once weekly. In previous trials, tirzepatide significantly reduced HbA1c and body weight, with greater effects than dulaglutide or semaglutide [61,62]. Given these findings, a phase II trial compared the effect of 5, 10, and 15 mg tirzepatide, 1.5 mg dulaglutide, and the placebo on the biomarkers of NASH and fibrosis in patients with T2DM [18]. Tirzepatide significantly decreased ALT level (10, 15 mg) compared with 1.5 mg dulaglutide, and decreased keratin-18 (10 mg) and Pro-C3 (15 mg) compared with the placebo (all *p* < 0.05).

In a substudy of the randomized phase 3 trial (SURPASS-3), LFC measured using MRI-PDFF was reduced more extensively with tirzepatide (relative decrease of 29.8%, 47.1%, and 39.6%, with 5, 10, and 15 mg/week, respectively) than with insulin degludec (11.2%) [19]. Specifically, the reduction in LFC was positively associated with the reduction in body weight, SAT, and VAT, thereby suggesting that the decrease in weight and adipose tissue is a possible mechanism for improvement in LFC.

In addition, in the phase 3 SURMOUNT-1 trial, body weight reduction was greater with tirzepatide than with the placebo in non-diabetic overweight/obese patients; at week 72, the mean body weight reduction was 15.0%, 19.5%, and 20.9% with 5, 10, and 15 mg/week of tirzepatide, respectively, compared to 3.1% with the placebo [63]. The magnitude of weight loss with tirzepatide, which exceeded the proposed threshold of 10% from baseline for improvements in hepatic fibrosis, suggests its potential role in the management of patients with NASH in the near future.

## 6. Dual GLP-1 and Glucagon Receptor Agonist

Based on the role of glucagon in regulating glucose homeostasis, a dual-receptor agonist for GLP-1 and glucagon was developed [64]. A phase IIb study with 834 obese patients with uncontrolled T2DM demonstrated the greater reduction in HbA1c and body weight in the cotadutide group compared with the placebo at weeks 14 and 54 (all *p* < 0.001) [65]. At 54 weeks of treatment, greater decreases in fatty liver index were observed with 300 µg cotadutide (mean change −8.18%; 95% CI, −9.79 to −6.57; *p* < 0.001) and 1.8 mg liraglutide (−6.22%; 95% CI, −8.38 to −4.06; *p* = 0.005) than with the placebo (−1.62%; 95% CI, −3.94 to 0.71). The decrease in FIB-4 with 300 µg cotadutide (−0.11; 95% CI, −0.17 to −0.05) was significantly greater than that of the placebo (0.05; 95% CI, −0.04 to 0.14; *p* = 0.004); however, this was comparable with that of liraglutide (−0.06; 95% CI, −0.15 to 0.02; *p* = 0.398). In addition, the reductions of AST levels in 300 µg cotadutide (−9.14%; 95% CI, −14.26 to −4.03) were significantly greater than those in liraglutide (0.35%; 95% CI, −6.54 to 7.24; *p* = 0.030) or in the placebo (5.65%; 95% CI, −1.69 to 12.98; *p* = 0.001). In 300 µg cotadutide group, the decrease in ALT levels were significantly greater (−14.15%; 95% CI, −19.75 to −8.55) than those in the liraglutide (−3.21%; 95% CI, −10.76 to 4.33; *p* = 0.023) or the placebo (0.93%; 95% CI, −7.10 to 8.97; *p* = 0.003).

## 7. The Effect of GLP-1RA Treatment in the Presence of T2DM

Previous studies have investigated the effect of GLP-1RA treatment in patients with and without T2DM. Notably, differences in the effect of treatment between the two groups have been observed in several studies. The LEAN trial included patients with and without T2DM, and showed a greater beneficial effect for the liraglutide treatment of NASH resolution, without worsening of fibrosis in patients with T2DM (RR = 4.7; 95% CI, 0.3–75.0), than those without (RR = 3.4; 95% CI, 0.8–14.4) [12]. In studies which evaluated the effect of liraglutide on LFC measured using MRI-PDFF, liraglutide (1.8 mg once daily for 26 weeks) achieved a greater decrease in LFC than insulin glargine in patients with NAFLD and T2DM. In contrast, the effect of liraglutide (3.0 mg once daily for 26 weeks) on LFC was comparable with that of the placebo in NAFLD patients without T2DM [38,44].

Given that the designs of GLP-1RA studies vary concerning dose, duration, and assessment modality (e.g., histology vs. MRI), comparison of the effect of GLP-1RA treatment according to the presence or absence of T2DM is insufficient. Although the underlying mechanism for the difference between patients with NAFLD with or without T2DM is unclear, patients with T2DM may have a greater degree of insulin resistance and/or less incretin effect than those without T2DM. Therefore, patients with NAFLD and concomitant T2DM could benefit more from GLP-1RAs treatment than those without T2DM. Future studies comparing the effect of GLP-1RA treatment according to the presence of T2DM are required to define optimal indications.

## 8. Ongoing Clinical Trials

Several ongoing clinical trials aim to evaluate the effect of GLP-1RAs treatment with paired histological evaluation (Table 3). A phase III trial of 1200 patients with biopsy-confirmed NASH and fibrosis stage 2 or 3 is ongoing, to investigate the efficacy and safety of a once-weekly semaglutide treatment compared with the placebo for 240 weeks (ESSENCE, NCT04822181). In a phase IV trial for 48 weeks, the effects of combined therapy with empaglifozin and semaglutide, empaglifozin monotherapy, and a placebo will be compared with 192 patients with biopsy-confirmed NASH with cirrhosis and T2DM (COMBATT2NASH, NCT04639414). Another ongoing phase II study is assessing the efficacy and safety of semaglutide monotherapy and combination regimens with cilofexor and/or firsocostat in 440 patients with biopsy-confirmed NASH-related compensated cirrhosis (NCT04971785). Given the favorable results from previous studies, a 52-week, multicenter, placebo-controlled, phase II RCT investigating the efficacy and safety of tirzepatide (5, 10, 15 mg/week) in overweight/obese patients with biopsy-confirmed NASH with fibrosis stage 2 or 3 is currently in progress (SYNERGY-NASH; NCT04166773). Finally, an 84-week, phase IIb/III RCT is in progress to investigate the efficacy and safety of cotadutide in patients with NASH and fibrosis stage 2 or 3 (PROXYMO-ADV, NCT05364931).

## 9. Conclusions

GLP-1RAs have not yet been approved for the treatment of NAFLD per se; however, the favorable results of recent studies have raised expectations for their role in the treatment of NAFLD, especially in patients with T2DM and/or obesity. However, there are insufficient data regarding the association of GLP-1RA treatment and liver-related mortality or morbidity. Another concern is the commonly encountered gastrointestinal adverse events during treatment with GLP-1RAs, albethey of mild-to-moderate severity in most cases, which may limit the tolerability. Forthcoming results from ongoing trials of once-weekly GLP-1RAs in NAFLD are expected to prove their safety and efficacy on fibrosis and NASH resolution.

## Figures and Tables

**Figure 1 ijms-24-09324-f001:**
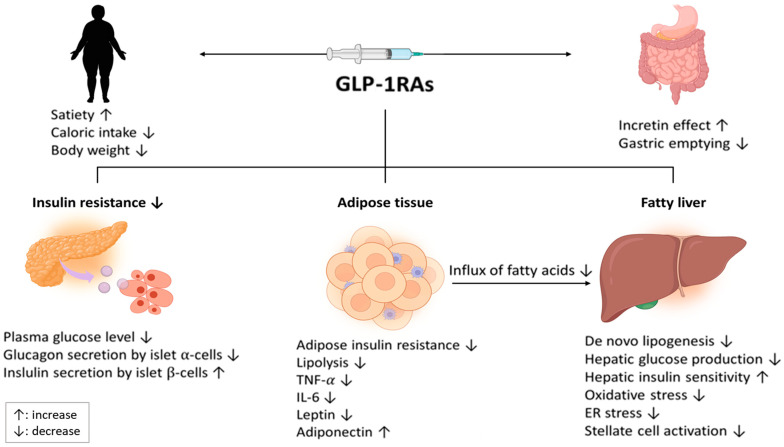
Therapeutic mechanisms of GLP-1RA treatment in patients with NAFLD. Abbreviations: GLP-1RA, Glucagon-like peptide-1 receptor agonist; NAFLD, nonalcoholic fatty liver disease; TNF, tumor necrosis factor, ER, endoplasmic reticulum.

**Figure 2 ijms-24-09324-f002:**
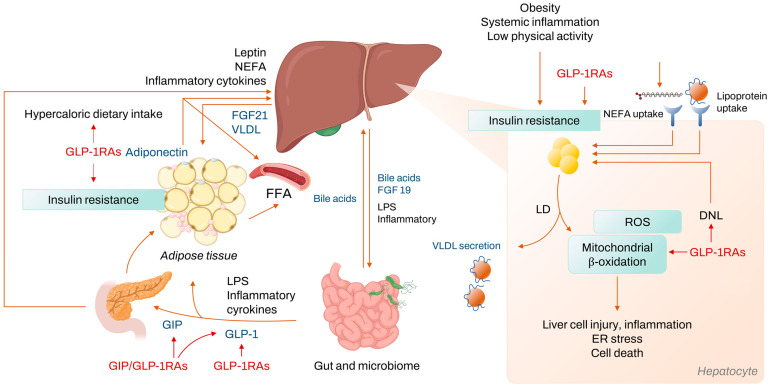
Mechanism of NASH development and treatment target of GLP-1RAs. Abbreviations: NASH, nonalcoholic steatohepatitis; GLP-1RA, Glucagon-like peptide-1 receptor agonist; NEFA, nonesterified fatty acids; FGF, fibroblast growth factor; VLDL, very low density lipoprotein; FFA, free fatty acid; LPS, lipopolysaccharide; GIP, glucose dependent insulinotropic peptide; ROS, reactive oxygen species; ER, endoplasmic reticulum; LD, lipid droplet; DNL, de novo lipogenesis.

**Table 1 ijms-24-09324-t001:** Characteristics of GLP-1 RAs that have been studied for the treatment of NAFLD.

GLP-1RA	First Approved	Molecular Weight (Da)	Reference Amino Acid Sequence	Included Components	Elimination Half-Life	Administration Schedule
**Short-acting compounds**						
Exenatide	2005	4186.6	Exendin-4	None	3.3–4.0 h	Twice daily
**Long-acting compounds**						
Liraglutide	2009	3751.2	Mammalian GLP-1	Free fatty acid *	12.6–14.3 h	Once daily
Dulaglutide	2014	59,670.6	Mammalian GLP-1	Immunoglobulin Fc fragment	4.7–5.5 day	Once weekly
Semaglutide	2017	4113.6	Mammalian GLP-1	Free fatty acid *	5.7–6.7 day	Once weekly

* Promoting binding to albumin. GLP-1RA, Glucagon-like peptide-1 receptor agonist; NAFLD, nonalcoholic fatty liver disease.

**Table 2 ijms-24-09324-t002:** Summary of clinical studies of GLP-1RAs treatment in NAFLD.

Author	Population	Comparators	Duration	Assessment	Findings	Safety (GLP-1RAs)
Armstrong et al. (UK) [12]	Overweight NASH	Liraglutide 1.8 mg/day (n = 26) Placebo (n = 26)	48 weeks	Histology	NASH resolution with no worsening of fibrosis: 39% with liraglutide vs. 9% with placebo (*p* = 0.019) Progression of fibrosis: 9% with liraglutide vs. 36% with placebo (*p* = 0.04)	SAE in 8% and treatment withdrawl due to AE in 8% of patients with liraglutide
Yan et al. (China) [38]	T2DM and NAFLD	Liraglutide 1.8 mg/day (n = 24) Sitagliptin 100 mg/day (n = 27) Insulin glargine 0.2 IU/kg/day (n = 24)	26 weeks	MRI-PDFF	Change in LFC: −4.0% with liraglutide vs. −3.8% with sitagliptin vs. −0.8% with insulin glargine (*p* = 0.911 for liraglutide vs. sitagliptin; *p* = 0.039 for liraglutide vs. insuline glargine; *p* = 0.043 for sitagliptin vs. insulin glargine)	AE in 20.8% and gastrointestinal disorders in 16.7% of patients with liraglutide
Bizino et al. (The Netherlands) [39]	T2DM with obesity or uncontrolled T2DM	Liraglutide 1.8 mg/day (n = 23) Placebo (n = 26)	26 weeks	MRI-PDFF	Change in LFC: 18.1% to 12.0% with liraglutide vs. 18.4% to 14.7% with placebo (estimated treatment effect −2.1% [95% CI −5.3 to 1.0])	Not reported
Feng et al. (China) [40]	T2DM and NAFLD	Liraglutide 1.8 mg/day (n = 31) Gliclazide 120 mg/day (n = 31) Metformin 2000 mg/day (n = 31)	24 weeks	Ultrasonography	Change in LFC: 36.7% to 13.1% with liraglutide vs. 33.0% to 19.6% with gliclazide vs. 35.1% to 18.4% with metformin (*p* < 0.01 for liraglutide vs. gliclazide)	Not reported
Guo et al. (China) [41]	Uncontrolled T2DM, obesity, and NAFLD	Insulin glargine (n = 30) Liraglutide 1.8 mg/day (n = 31) Placebo (n = 30)	26 weeks	H-MRS	Change in LFC: 26.4% to 20.6% with liraglutide (*p* < 0.05) vs. 25.0% to 22.6% with insulin glargine (*p* > 0.05)	Not reported
Zhang et al. (China) [42]	T2DM and NAFLD	Liraglutide 1.2 mg/day (n = 30) Pioglitazone 30 mg/day (n = 30)	24 weeks	H-MRS	Change in LFC: 24.1 to 20.1 with liraglutide vs. 23.9 to 22.4 with pioglitazone (*p* < 0.05)	AE in 33.3% and gastrointestinal reactions in 30% of patients with liraglutide
Bouchi et al. (Japan) [43]	T2DM with insulin treatment and obesity	Liraglutide 0.9 mg/day + insulin (n = 8) Insulin (n = 9)	36 weeks	CT	Change in liver attenuation index: 0.84 to 0.99 with liraglutide + insulin vs. 0.99 to 1.06 with insulin (*p* = 0.065)	No severe AE
Khoo et al. (Singapore) [44]	NAFLD and obesity without T2DM	Liraglutide 3 mg/day (n = 15) Moderate-intensity exercise (n = 15)	26 weeks	MRI	Change in LFC: −7.0 ± 7.1% with liraglutide vs. −8.1 ± 13.2% with exercise (*p* = 0.78) Change in liver stiffness: −0.25 ± 0.27 kPa with liraglutide vs. −0.12 ± 0.19 kPa with exercise (*p* = 0.17)	Nausea in 80%, abdominal discomfort in 100%, and diarrhea in 33% of patients with liraglutide
Nwesome et al. (UK) [13]	Biopsy-confirmed NASH and liver fibrosis of stage F1–3 and obesity	Semaglutide 0.1 mg (n = 80), 0.2 mg (n = 78), 0.4 mg/day (n = 82) Placebo (n = 80)	72 weeks	Histology	NASH resolution with no worsening of fibrosis: 40% with semaglutide 0.1 mg, 36% with 0.2 mg, 59% with 0.4 mg, and 17% with placebo (*p* < 0.001 for semaglutide 0.4 mg vs. placebo) Improvement in fibrosis stage: 43% of semaglutide 0.4 mg vs. 33% of placebo (*p* = 0.48).	Nausea in 40%, constipation in 22%, vomiting in 15%, and malignancy in 1% of patients with semaglutide
Volpe et al. (Italy) [45]	Uncontrolled T2DM and NAFLD	Semaglutide 0.5 mg/week (n = 40)	52 weeks	Ultrasonography	70% achieved at least one-class reduction in the 4-point semiquantitative staging (*p* < 0.001)	Not reported
Flint et al. (Multinational) [46]	NAFLD and obesity	Semaglutide 0.4 mg/day (n = 34) Placebo (n = 33)	48 weeks	MRI-PDFFMRE	≥30% reduction in LFC: 76.5% with semaglutide vs. 30.3% with placebo (estimated treatment ratio 0.47 [95% CI 0.36 to 0.60; *p* < 0.001]) ≥15% reduction in liver stiffness: 17.6% with semaglutide vs. 15.2% with placebo (etimated treatment ratio 0.96 [95% CI 0.89 to 1.03; *p* = 0.2798])	AE in 93.9%, SAE in 12.1%, drug discontinuation due to AE in 3.0% of patients with semaglutide
Loomba et al. (USA) [47]	Biopsy-confirmed NASH-related cirrhosis and BMI ≥27 kg/m^2^	Semaglutide 2.4 mg/week (n = 47) Placebo (n = 24)	48 weeks	Histology	Improvement in liver fibrosis of one stage or more without worsening of NASH: 11% with semaglutide vs. 29% with placebo (OR 0.28 [95% CI 0.06 to 1.24; *p* = 0.087]) NASH resolution: 34% with semaglutide vs. 21% with placebo (OR 1.97 [95% CI 0.56 to 7.91; *p* = 0.29])	AE in 89%, SAE in 13%, nausea in 45%, diarrhea in 19%, and vomiting in 17% of patients with semaglutide
Liu et al. (China) [48]	Newly diagnosed T2DM and NAFLD	Exenatide 5 μg/10 μg bid (n = 38) Insulin glargine 0.1–0.3 IU/kg/day (n = 38)	24 weeks	H-MRS	Change in LFC: −17.55 ± 12.93% (*p* < 0.05) with exenatide vs. −10.49 ± 11.38% (*p* < 0.05) with insulin glargine	AE in 13.16% and hypoglycemia in 7.89% of patients with exenatide
Dutour, et al. (France) [49]	Uncontrolled T2DM and obesity	Exenatide 5 μg/10 μg bid (n = 22) Control (n = 22)	26 weeks	H-MRS	Change in LFC: −23.8 ± 9.5% with exenatide vs. +12.5 ± 9.6% with control (*p* = 0.007)	Not reported
Shao et al. (China) [50]	Newly diagnosed obesity, T2DM, and NAFLD	Exenatide + Insulin glargine (n = 30) Insulin (n = 30)	12 weeks	Ultrasonography	Reversal rate of fatty liver: 93.3% with exenatide vs. 66.7% with insulin (*p* < 0.01)	Not reported
Kuchay et al. (India) [51]	T2DM and NAFLD	Dulaglutide 1.5 mg/week (n = 27) Control (n = 25)	24 weeks	MRI-PDFF VCTE	Change in LFC: −5.8 ± 1.0% with dulaglutide vs. −2.3 ± 1.2% with control (between-group difference −3.5% [95% CI −6.6 to −0.4; *p* = 0.025]) Change in liver stiffness: −1.43 ± 0.56% with dulaglutide vs. −0.12 ± 0.63% with control (between-group difference −1.31% [95% CI −2.99 to 0.37; *p* = 0.123])	Three discontinued due to upper gastrointestinal upset and no SAE in patients with dulaglutide
Seko et al. (Japan) [52]	Biopsy-proven NAFLD with T2DM	Dulaglutide 0.75 mg/week (n = 15)	12 weeks	Controlled attenuation parameter, VCTE	Change in LFC: 313.6 to 333.4 dB/m (*p* = 0.080) Change in liver stiffness: 9.3 to 6.9 kPa (*p* = 0.043)	One with diarrhea
Gastaldelli et al. (Italy) [19]	Uncontrolled T2DM and obesity	Tirzepatide 5 mg (n = 71); 10 mg (n = 79); 15 mg (n = 72) Insulin degludec (n = 74)	52 weeks	MRI-PDFF	Change in LFC: –8.09 ± 0.57% with pooled tirzepatide 10 mg and 15 mg vs. –3.38 ± 0.83% with insulin degludec group (estimated treatment difference –4.71% [95% CI –6.72 to –2.70; *p* < 0.0001])	One discontinued due to adverse event

GLP-1RA, Glucagon-like peptide-1 receptor agonist; NAFLD, nonalcoholic fatty liver disease; NASH, nonalcoholic steatohepatitis; AE, adverse event; SAE, serious adverse event; T2DM, type 2 diabetes mellitus; MRI, magnetic resonance imaging; PDFF, proton density fat fraction; LFC, liver fat content; CI, confidence interval; MRS, MR spectroscopy; CT, computed tomography; BMI, body mass index; OR, odds ratio; VCTE, vibration-controlled transient elastography.

**Table 3 ijms-24-09324-t003:** Ongoing clinical studies of GLP-1RAs treatment in NAFLD.

Name	Phase	Population	Comparators	Duration	Assessment	Primary Endpoints
ESSENCE, NCT04822181	III	Biopsy-confirmed NASH with fibrosis stage 2 and 3 (n = 1200)	Semaglutide once weekly Placebo	72 weeks	Histology	Resolution of steatohepatitis and no worsening of liver fibrosis Improvement in liver fibrosis and no worsening of steatohepatitis Time to first liver-related clinical event
COMBATT2NASH, NCT04639414	IV	T2DM and biopsy-confirmed NASH with fibrosis stage 1–3 (n = 192)	Combined treatment with Empagliflozin once daily and Semaglutide once weekly Empagliflozin once daily monotherapy Placebo	48 weeks	Histology	Histological resolution of NASH without worsening of fibrosis
NCT04971785	II	Biopsy confirmed NASH-related cirrhosis (n = 440)	Semaglutide once weekly + cilofexor and firsocostat Semaglutide once weekly monotherapy Cilofexor and firsocostat monotherapy Placebo	72 weeks	Histology	Percentage of participants who achieve ≥1-stage improvement in fibrosis without worsening of NASH Percentage of participants with NASH resolution
NCT03884075	II	Histological or imaging evidence of hepatic steatosis (n = 84)	Semaglutide once weekly No intervention	30 weeks	Histology MRS	≥1 point decrease in NAFLD activity score Reduction in liver fat content ≥25% and reduction in ALT by ≥25% or normalization of ALT
NCT05067621	III	Impaired glucose tolerance or T2DM and NAFLD (n = 60)	Semaglutide once weekly Placebo	24 weeks	MRI-PDFF	Change in oral disposition index Change in MRI-PDFF
NCT05016882	II	Biopsy-confirmed NASH with fibrosis stage 2–4 (n = 672)	NNC0194 0499 once weekly + semaglutide once weekly Semaglutide once weekly monotherapy Placebo	52 weeks	Histology	Improvement in liver fibrosis and no worsening of NASH
SYNERGY-NASH; NCT04166773	II	T2DM and biopsy-confirmed NASH with fibrosis stage 2–3, and obesity (n = 196)	Tirzepatide once weekly Placebo	52 weeks	Histology	Percentage of participants with absence of NASH with no worsening of fibrosis
NCT05751720	I/II	T2DM and NAFLD with fibrosis stage 3 and 4 (n = 30)	Tirzepatide once weekly	12 months	VCTE, MRI-PDFF	Change in liver stiffness in terms of kPa Change in liver fat quantification
PROXYMO-ADV, NCT05364931	IIb/III	Biopsy-confirmed NASH with fibrosis stage 2 and 3 (n = 45)	Cotadutide once daily Placebo	84 weeks	Histology	Proportion of participants with resolution of NASH without worsening of liver fibrosis Proportion of participants with improvement in liver fibrosis by at least one stage without worsening of NASH

GLP-1RA, glucagon-like peptide-1 receptor agonist; NAFLD, nonalcoholic fatty liver disease; NASH, nonalcoholic steatohepatitis; T2DM, type 2 diabetes mellitus; MRI, magnetic resonance imaging; PDFF, proton density fat fraction; MRS, MR spectroscopy; VCTE, vibration-controlled transient elastography.

## Data Availability

Not applicable.

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
