# Peer review of "Therapeutic Mechanisms and Clinical Effects of Glucagon-like Peptide 1 Receptor Agonists in Nonalcoholic Fatty Liver Disease"

_ijms, 2023, doi:10.3390/ijms24119324_

Round 1
Reviewer 1 Report
Manuscript Number: ijms-2381956
The manuscript entitled “ Therapeutic Mechanisms and Clinical Effects of Glucagon-Like 2 Peptide 1 Receptor Agonists in Nonalcoholic Fatty Liver Dis-3 ease” is based upon a very good review. This review report is very interesting, well-written, and suitable for this journal.
Comments
- Add one or two sentences related to the connection between non-alcoholic fatty liver diseases and Glucagon-like peptide-1 receptor agonists
- You can also mention how the Glucagon-like peptide-1 receptors play a role in hepatosteatosis through the cross-talk between the liver and adipose tissue.
- In Figure 1 you can increase the font size of the letters
- In line 77, “In preclinical models of NASH, administration of AC3174, an analogue of exenatide, 76 led to significant reductions in body weight, LFC, plasma alanine aminotransferase (ALT) 77 levels, and lipids concentration” [24]. Reduction in lipid concentrations means both TG and cholesterol in plasma or in the hepatic tissue?
- Treatment Mechanisms of GLP-1RAs on NAFLD is well written and incorporate molecular connection of GLP-1RAs and lipogenesis
- In Table 1, keep the same alignment and what is the reason for mentioning clinical studies done places in the Table?
- You can make a graphical representation of NASAH development and different treatment strategies. Then highlight your point of interest GLP-1RAs.
- In lane 196 “A higher percentage of patients….” Add appropriate reference
- Here you mentioned different drugs, make a diagram to illustrate their mode of action and pathways accordingly.
- Table 2 is also not in a good format, correct the alignment
- Appreciated for including the most relevant references
Author Response
1. Add one or two sentences related to the connection between non-alcoholic fatty liver diseases and Glucagon-like peptide-1 receptor agonists
Response) Thank you for your kind comment. We added a sentence "The role of GLP-1RAs on NAFLD could be also clarified by its effects on body weight re-duction, oxidative stress, and the gut-liver axis." to the "Introduction" section.
2. You can also mention how the Glucagon-like peptide-1 receptors play a role in hepatosteatosis through the cross-talk between the liver and adipose tissue.
Response) Thank you for your kind comment. We added a paragraph regarding the relation between liver and adipose tissue in "Treatment Mechanisms of GLP-1RAs on NAFLD"
3. In Figure 1 you can increase the font size of the letters
Response) Thank you for your kind comment. We increased the size of letters.
4. In line 77, “In preclinical models of NASH, administration of AC3174, an analogue of exenatide, 76 led to significant reductions in body weight, LFC, plasma alanine aminotransferase (ALT) 77 levels, and lipids concentration” [24]. Reduction in lipid concentrations means both TG and cholesterol in plasma or in the hepatic tissue?
Response) Thank you for your precise comment. We changed the sentence as follows :In preclinical models of NASH, administration of AC3174, an analogue of ex-enatide, led to significant reductions in body weight, LFC, and levels of plasma alanine aminotransferase (ALT) and triglyceride
5. Treatment Mechanisms of GLP-1RAs on NAFLD is well written and incorporate molecular connection of GLP-1RAs and lipogenesis
Response) Thank you for your kind comment. We revised the paragraph of lipogenesis in "Treatment Mechanisms of GLP-1RAs on NAFLD" as follows: In addition, hyperglycemia activates the carbohydrate-responsive element-binding protein (ChREBP) transcriptional factor, a key modulator of de novo liver lipogenesis, suggesting that GLP-1RAs could reduce ChREBP activation and, consequently, decrease de novo liver lipogenesis and LFC [32]. Potential benefits of GLP-1RAs in lipid metabolism have been demonstrated in experimental studies; GLP-1RAs reduced very low-density lipoprotein (VLDL) production and the hepatic expression of genes involved in VLDL production and lipogenesis, such as SREBP-1c and fatty acid synthase [33]. This evidence confirms that GLP-1RAs reduce the accumulation of hepatic triglycerides by limiting hepatic lipogenesis.
6. In Table 1, keep the same alignment and what is the reason for mentioning clinical studies done places in the Table?
Response) Thank you for your kind comment. We revised the table following your comment. We created this table to compare the design of the study and the effects of GLP-1RAs
7. You can make a graphical representation of NASAH development and different treatment strategies. Then highlight your point of interest GLP-1RAs.
Response) Thank you for your kind comment, however, due to the time limitation, we cannot make a new graphic of NASH develepmont. We are sorry for this. If you and editorial office can give us a couple of weeks, we can develop it.
8. In lane 196 “A higher percentage of patients….” Add appropriate reference
Response)
Thank you for your kind comment. We added reference 12.
9. Here you mentioned different drugs, make a diagram to illustrate their mode of action and pathways accordingly.
Response) Thank you for your kind comment, however, due to the time limitation, we cannot make a new graphic of NASH development. We are sorry for this. We can develop it if you and editorial office can give us a couple of weeks, we. Table 2 is also not in a good format, correct the alignment
Response) Thank you for your kind comment. We revised the table following your comment.
11. Appreciated for including the most relevant references
Response) Thank you for your generous comment.
Reviewer 2 Report
The paper is a review of the Therapeutic mechanism and ongoing studies with Glucagon-like Peptide 1 Receptor agonists in Non-alcoholic Fatty liver disease. This review is very important to understand the complexity of this disease and possible future treatments. Overall, the review is very interesting, showing all studies that have been done with GLP-1RA agonists and ongoing clinical studies. The literature review is excellent and seems adequate. However, I missed a figure that explains better the signaling mechanism of GLP-1RA. Which kinases GLP-1RA activates in the pancreas? This should be the first figure of the review. I think this information is important in a review and a figure helps to elucidate. Besides that, the review is excellent.
Author Response
Response) Thank you for your kind comment; however, due to the time limitation, we cannot make a new graphic of signaling mechanism of GLP-1RAs. We are sorry for this. We can develop it if you and the editorial office can give us a couple of weeks.
Reviewer 3 Report
This is an interesting review collective describing the therapeutic role of GLP-1RAs in NAFLD conditions. The authors have done a good job of tabulating the findings from different clinical studies.
Although the mechanism of the drug is well explained, the authors could add a brief explanation of the differences between the different GLP-1RAs (How does liraglutide differ from semaglutide), the clinical data of which has been discussed in the second half of the paper individually. This would help in better understanding the topic from a non-medical perspective. A schematic could also suffice.
Author Response
Response) Thank you for your kind comment; however, due to the time limitation, we cannot make a new graphic of NASH development. We are sorry for this. We can develop it if you and the editorial office can give us a couple of weeks.
We added the Table 1 of "Characteristics of GLP-1 RAs that have been studied for the treatment of NAFLD" following your comment.